# DiffRhythm 2: Efficient and High Fidelity Song Generation via Block Flow Matching

## Abstract

Generating full-length, high-quality songs is challenging, as it requires maintaining long-term coherence both across text and music modalities and within the music modality itself. Existing non-autoregressive (NAR) frameworks, while capable of producing high-quality songs, often struggle with the alignment between lyrics and vocal. Concurrently, catering to diverse musical preferences necessitates reinforcement learning from human feedback (RLHF). However, existing methods often rely on merging multiple models during multi-preference optimization, which results in significant performance degradation. To address these challenges, we introduce DiffRhythm 2, an end-to-end framework designed for high-fidelity, controllable song generation. To tackle the lyric alignment problem, DiffRhythm 2 employs a semi-autoregressive architecture based on block flow matching. This design enables faithful alignment of lyrics to singing vocals without relying on external labels and constraints, all while preserving the high generation quality and efficiency of NAR models. To make this framework computationally tractable for long sequences, we implement a music variational autoencoder (VAE) that achieves a low frame rate of 5 Hz while still enabling high-fidelity audio reconstruction. In addition, to overcome the limitations of multi-preference optimization in RLHF, we propose cross-pair preference optimization. This method effectively mitigates the performance drop typically associated with model merging, allowing for more robust optimization across diverse human preferences. We further enhance musicality and structural coherence by introducing stochastic block representation alignment loss. Experimental results demonstrate that DiffRhythm 2 can generate complete songs up to 210 seconds in length, consistently outperforming existing open-source models in both subjective and objective evaluations, while maintaining efficient generation speed. To encourage reproducibility and further exploration, we will release the inference code and model checkpoints. Audio samples are available at: https://anonymouspapercc.github.io/diffrhythm2/

## 1 Introduction

Music serves as an abstract expression of human emotion and culture. Songs represent a unique art form, combining music and language to convey emotions and stories through melody and lyrics. With the rapid development of deep learning, music generation has progressed beyond symbolic music generation (Wang et al., 2025; Qu et al., 2024) and singing voice synthesis (Zhang et al., 2022; Chen et al., 2020; Ning et al., 2025b), moving towards more challenging tasks such as instrumental music generation (Copet et al., 2023; Schneider et al., 2023; Zhang et al., 2025) and song generation. Unlike singing voice synthesis, which produces vocals with predefined melodies, or instrumental music generation, which only models melody and instrumentation, song generation requires joint modeling of lyrics, structure, singing vocal and accompaniment. This is further complicated by the typical length of songs, which often exceeds three minutes, making lyric alignment and stable style modeling particularly challenging.

Early approaches, such as Jukebox (Dhariwal et al., 2020) and SongCreator (Lei et al., 2024), generate songs by predicting the acoustic tokens of vocals and accompaniment mixed into a single track. However, the limited information density of these tokens hampers high-quality song generation. Yue (Yuan et al., 2025) improves generation quality by modeling the vocals and accompaniment on separate tracks, while SongGen (Liu et al., 2025) further enhances quality by introducing interleaved

dual-track prediction. Despite these improvements, Yue and SongGen predict the tracks independently, which often leads to inconsistencies between the vocals and accompaniment. LeVo (Lei et al., 2025) addresses this issue by first generating a mixed track and using it to guide the separate prediction of vocals and accompaniment, thereby reducing mismatches. However, the overall quality is still limited by the information compression in the mixed track tokenization, and it is difficult to generate complex and rich accompaniment. Most of these methods are based on autoregressive frameworks, which result in slow generation speeds. This hinders real-time applications and user interactivity.

Unlike the aforementioned approaches, DiffRhythm (Ning et al., 2025a) takes a bold approach within a non-autoregressive (NAR) framework. By leveraging the generation ability of diffusion models and continuous representations, it achieves high-quality mixed-track generation of vocals and accompaniment at over 50 times faster than autoregressive methods. However, NAR models struggle with lyric alignment in long sequences. DiffRhythm addresses this issue by conditioning on sentence-level timestamps. However, this solution significantly reduces creativity and diversity, and also imposes higher requirements on the training data. ACE-Step (Gong et al., 2025) solved the alignment problem without timestamps by introducing representation alignment (REPA) (Yu et al., 2024) loss with mHuBERT (Boito et al., 2024) as semantic constraints. However, the addition of these constraints significantly reduced musicality, creating a delicate trade-off between lyric alignment and generation quality.

In addition, pretrained models often fail to align with human preferences due to the gap between training objectives and desired song quality. Therefore, a typical method is to introduce reinforcement learning from human feedback (RLHF) in the post-training process to refine generation preferences across various musical dimensions. DiffRhythm+ (Chen et al., 2025) demonstrated the feasibility of improving overall musicality with Direct Preference Optimization (DPO) (Rafailov et al., 2023). LeVo extended this approach by applying DPO to different dimensions separately and interpolating model weights to enhance overall performance. While this strategy improved balance across dimensions, it inevitably limited the upper bound of each individual capability due to the averaging effect.

To address these challenges, we present DiffRhythm 2, a semi-autoregressive end-to-end framework for song generation based on block flow matching. This framework partitions latent representations into fixed-length blocks and generates each block non-autoregressively using flow matching, while maintaining autoregressive dependencies between blocks. This design enables faithful lyric alignment without requiring additional labels or constraints. The semi-autoregressive structure provides rich bidirectional context within blocks, ensuring long-sequence consistency, and supports fast inference for songs up to 210 seconds long. To accommodate block flow matching during training, we introduce stochastic block REPA loss that ensures efficient training and provides accurate representation guidance for the current block. We group similar preferences and perform pairwise optimization across groups to improve the performance of multi-preference alignment. Finally, as the block flow matching training strategy significantly increases the sequence length, we implement a high-compression music variational autoencoder (VAE), achieving high compression while still enabling high-quality audio reconstruction. To encourage reproducibility and further exploration, we will release the inference code and model checkpoints. The main contributions of this paper are summarized below:

- We propose DiffRhythm 2, a novel semi-autoregressive song generation framework based on block flow matching, which is capable of producing high-quality songs with faithful lyric alignment and excellent accompaniment performance.

- We introduce stochastic block REPA loss to enable representation alignment in block flow matching, thereby enhancing structural modeling and achieving remarkable coherence and hierarchical structure in long-form song generation.

- We propose a cross-pair preference optimization strategy to efficiently handle multi-preference alignment, which enhances final performance while reducing the number of optimized models through a group-based optimization approach.

- We design a music VAE with low-frame-rate compression at 5 Hz that preserves reconstruction quality while reducing sequence length, which not only accelerates inference but also makes long-sequence modeling in DiffRhythm 2 both feasible and effective.

## 2 RELATED WORKS

### 2.1 FLOW MATCHING AND AUTOREGRESSIVE DIFFUSION MODEL

Flow matching (FM) (Lipman et al., 2023) trains continuous normalizing flows (CNFs) without simulation by directly regressing a time-dependent vector field $v_t(x)$ that transports samples from a simple prior $p_0$ to a target distribution $p_1$ along paths $p_t$. This yields simpler objectives, faster convergence, and more stable training than score matching. Conditional flow matching (CFM) further conditions the vector field on external information, enabling guided generation. CFM is widely used in the field of image generation and has also been applied to TTS tasks in recent years.

Autoregressive diffusion models aim to combine the strengths of both autoregressive and non-autoregressive paradigms, achieving complementary advantages. These approaches can generally be categorized into two groups: diffusion-backbone and language-model-backbone designs. For diffusion-based backbones, the SSD series (Han et al., 2022; 2023) introduced this idea and applied it to text generation tasks, with Block Diffusion further optimizing the framework. ARDiT (Liu et al., 2024) extended this line of work to text-to-speech (TTS). On the other hand, language-model-based backbones include DiTAR (Jia et al., 2025), which integrates diffusion mechanisms into autoregressive language models.

### 2.2 SONG GENERATION

Song generation aims to produce coherent vocals and accompaniment given lyrics and style specifications, requiring joint modeling of melody, harmony, and lyric alignment. Early methods adopted sequential pipelines: Melodist (Hong et al., 2024) and MelodyLM (Li et al., 2024) first generated vocals from lyrics, then synthesized accompaniment. However, since vocals and accompaniment are inherently interdependent, this sequential approach often produces suboptimal results. To address these limitations, parallel generation methods emerged. SongCreator (Lei et al., 2024) and Yue (Yuan et al., 2025) generate vocals and accompaniment independently in parallel, while SongGen (Liu et al., 2025) introduces interleaved prediction across tracks to improve quality. Despite these advances, weak coupling between vocal and instrumental tracks often leads to harmonic inconsistencies. LeVo (Lei et al., 2025) mitigates this by first generating a mixed track, then using it to guide separate vocal and accompaniment synthesis, achieving better harmony at the cost of reduced accompaniment complexity. Recent work has explored alternative architectures to improve both quality and efficiency. MusicCoT (Lam et al., 2025) and Songbloom (Yang et al., 2025) leverage Chain-of-Thought reasoning (Wei et al., 2022) to enhance vocal-accompaniment coordination and overall generation quality. However, these language-model-based approaches suffer from computational overhead and struggle with style consistency across long sequences due to their autoregressive nature. Non-autoregressive approaches offer promising alternatives. DiffRhythm (Ning et al., 2025a) and ACE-Step (Gong et al., 2025) employ diffusion models to achieve superior long-sequence consistency and vocal-accompaniment harmony while maintaining faster inference. However, both methods face challenges in lyric alignment for extended sequences. While each proposes specific solutions, these fixes come at the cost of reduced creativity or musicality. Specifically, DiffRhythm relies on timestamp conditioning while ACE-Step employs representation alignment, highlighting the need for more sophisticated alignment strategies.

## 3 METHOD

### 3.1 OVERVIEW

The overall architecture of DiffRhythm 2, illustrated in Figure 1, comprises of a Music VAE and a Diffusion Transformer enhanced with block flow matching. The Diffusion Transformer is conditioned on lyrics $L$ together with either text style prompt $S_t$ or audio style prompt $S_a$, and generates VAE latents block by block. These latents are then passed to the Music VAE decoder to reconstruct the waveform. For training, in addition to the block flow matching loss, we introduce the stochastic block REPA loss combined with MuQ (Zhu et al., 2025) representations to enhance the model's musicality and structural modeling. Moreover, to address the degraded average performance caused by multi-preference optimization, we introduce a cross-pair preference optimization strategy. The

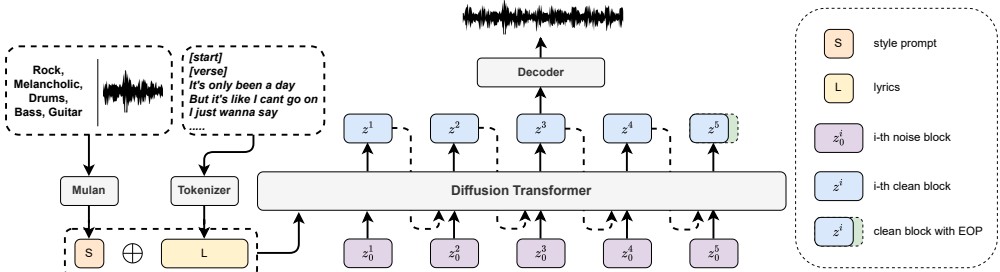

Figure 1: Overview architecture of our proposed DiffRhythm 2. Either text description or audio can specify the style prompt.

following subsections will provide detailed descriptions of the DiffRhythm 2 modules and training procedure.

## 3.2 MUSIC VAE

To address the challenge of training DiffRhythm 2 on extremely long sequences, we design a customized Music VAE with a frame rate as low as 5 Hz. The VAE processes 24 kHz input audio as input and reconstructs it at 48 kHz, achieving compression ratios of 4800× during encoding and 9600× during decoding. The Music VAE consists of an encoder, a transformer block, and a decoder. The encoder adopts the same architecture as Stable Audio 2 VAE [1](Evans et al., 2025). A transformer block is inserted before the decoder to alleviate reconstruction pressure. To maximize reconstruction quality, we employ BigVGAN (Lee et al., 2023) as the decoder. For the training loss, we combine the multi-scale mel loss from BigVGAN with the multi-scale STFT loss from Stable Audio 2 VAE to jointly optimize vocals and accompaniment. In addition, we employ a set of discriminators, including the multi-period discriminator, multi-scale discriminator, and CQT discriminator from BigVGAN.

Unlike MIMI (Défossez et al., 2024), we deliberately avoid adding a similar transformer block after the encoder for two reasons: (1) The limited future information introduced by convolution helps enhance content continuity between consecutive blocks. (2) The global bidirectional receptive field of transformer introduces excessive future information, substantially increasing the burden on the generative model. We explore the reconstruction performance of our VAE in Appendix A.

## 3.3 BLOCK FLOW MATCHING

### 3.3.1 DEFINITION

We first establish the mathematical foundation of flow matching before extending it to the block-wise setting. Given the target latent $Z$, flow matching defines a linear probability path that transports Gaussian noise $Z_0 \sim \mathcal{N}(0, I)$ to the target data distribution $Z$. During training, the timestep $t$ is sampled from $\mathcal{U}[0, 1]$, and the noisy latent $Z_t$ is obtained by linear interpolation:

$$Z_t = (1 - t)Z_0 + tZ. \tag{1}$$

The model $f_\theta$ conditions on $Z_t$, style prompt $S$, lyrics $L$, and timestep $t$ to estimate the velocity field $\hat{v}$. Since the path is linear, the ground truth velocity field $v$ is $Z - Z_0$. The flow matching loss is therefore defined as

$$\mathcal{L}_{fm} = \mathbb{E}\big[\|\hat{v} - v\|_2^2\big] = \mathbb{E}\big[\|f_\theta(S, L, Z_t, t) - (Z - Z_0)\|_2^2\big]. \tag{2}$$

In block flow matching, the core idea is that each block is generated with flow matching, while the dependency across blocks is handled autoregressively. Let the block size be $b$, and split the target latent $Z$ of length $l$ into $Z = \{z^1, z^2, \cdots z^k\}$, where $k = \lceil l/b \rceil$ denotes the total number of blocks.

---

[1]https://github.com/Stability-AI/stable-audio-tools

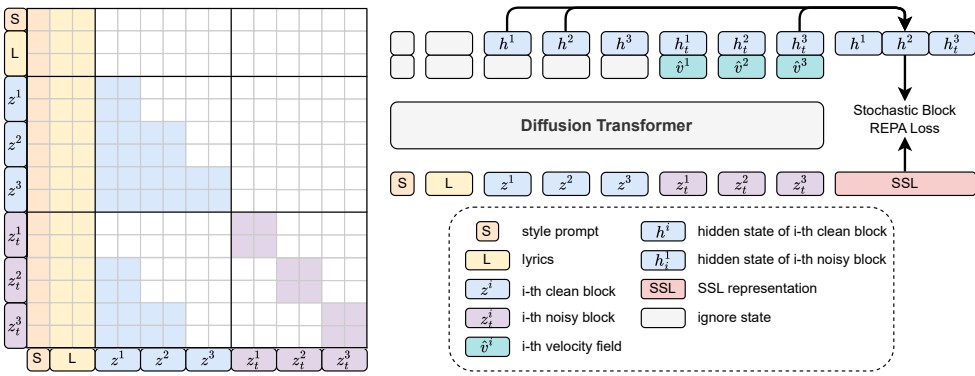

Figure 2: The right panel shows the block-level latent sequence structure of the inputs and outputs, while the left panel shows the corresponding attention mask applied during training. Note that the block size here is 2.

Since the generation of each block depends on all previously generated blocks, the velocity field for the $i$-th block is estimated as

$$\hat{v}^i = f_\theta(S, L, z^{<i}, z_t^i, t^i), \tag{3}$$

where $z^{<i}$ represents all preceding blocks. $z_t^i$ is the noisy latent for the current block obtained via linear interpolation, and $t^i \sim \mathcal{U}[0,1]$ is sampled independently for each block. The corresponding ground truth velocity for the i-th block is $v^i = z^i - z_0^i$, where $z_0^i \sim \mathcal{N}(0, I)$ is the standard Gaussian noise for this block. The overall block flow matching loss is then defined as

$$\mathcal{L}_{bfm} = \frac{1}{k}\sum_{i=1}^{k}\mathbb{E}\big[\|\hat{v}^i - v^i\|_2^2\big] = \frac{1}{k}\sum_{i=1}^{k}\mathbb{E}\big[\|f_\theta(S, L, z^{<i}, z_t^i, t^i) - (z^i - z_0^i)\|_2^2\big]. \tag{4}$$

### 3.3.2 DETAILS OF APPLICATION

From the definition, block flow matching requires access to clean block. While this is naturally available during inference, the training input sequence typically consists only of $(S, L, Z_t)$ without the clean blocks. To address this, we concatenate the clean sequence to the noisy sequence, forming $(S, L, Z, Z_t)$, and apply an attention mask to enforce autoregressive dependencies. Figure 2 illustrates our attention mask design: the style prompt $S$ and lyrics $L$ can be attended by any block; for the clean sequence, the $i$-th block can only attend to blocks 1 through $i$; for the noisy sequence, the $i$-th block can only attend to clean blocks 1 through $i-1$ and its own noisy block. This design enforces autoregressive training but results in very long input sequences. To alleviate this, we employ a 5 Hz music VAE to substantially compress the training sequence length.

The introduction of clean sequences requires a mechanism to distinguish them from noisy sequences to facilitate model training. In conventional autoregressive frameworks, this is typically achieved by inserting special tokens as delimiters. However, such an approach is not applicable here. Since the receptive field is controlled by the attention mask, inserting delimiters for each block is impractical and would also affect positional encoding. To address this in a simple yet effective way, we leverage the timestep $t$ to differentiate between sequences. Specifically, we set the style prompt and lyrics to a fixed timestep of $-1$, the clean sequence to 1, and assign the noisy sequence a timestep sampled from $\mathcal{U}[0,1]$. To further enhance training stability, different blocks within the same sequence are assigned independently sampled timesteps.

### 3.3.3 VARIABLE LENGTH GENERATION

Unlike prior works that predetermine the total sequence length before generation, DiffRhythm 2 incorporates the End-of-Prediction (EOP) frame as part of the prediction target to enable variable length generation. Since generation proceeds block by block, we pad $n$ EOP frames at the end of the sequence rather than a single frame. Specifically, $n$ is determined by the length of the final block $l^k$:

$$n = l^k \bmod b. \tag{5}$$

**Algorithm 1** Block Flow Matching Training

**Require:** forward noise process $q_t$
  **repeat**
    $t_{\text{noise}} \leftarrow \{ t^i \sim \mathcal{U}[0,1] \mid i = 1, \ldots, k \}$
    $Z_t \leftarrow q_{t_{noise}}(Z)$
    $\hat{v} \leftarrow f_\theta(S, L, Z, Z_t, t_{\text{noise}})$
    $\theta \leftarrow \theta - \eta \nabla_\theta \mathcal{L}_{\text{bfm}}(\hat{v})$
  **until** converged

**Algorithm 2** Block Flow Matching Inference

**Require:** max block number $N$, sampling process $SP$, KV cache $KV$, model with KV cache $m_\theta$
  $Z \leftarrow \emptyset$
  $\emptyset, KV \leftarrow m_\theta(S, L, \emptyset, -1, \emptyset)$
  **for** $i = 1$ **to** $N$ **do**
    $z_0^i \sim \mathcal{N}(0, I)$
    $z^i \leftarrow SP(m_\theta, z_0^i, KV)$
    $\emptyset, KV^i \leftarrow m_\theta(\emptyset, \emptyset, z^i, 1, KV)$
    $KV \leftarrow KV \oplus KV^i$
    $Z \leftarrow Z \oplus z^i$
    **if** $EOP \in z^i$ **then**
      **break**
    **end if**
  **end for**
  **return** $Z$

If $l^k$ is shorter than the block size $b$, EOPs are padded to complete the block; if $l^k$ equals $b$, an additional full block of EOPs is padded. Formally,

$$Z = \text{concat}(Z_{\text{orig}}, \text{repeat}(\text{EOP}, n)). \tag{6}$$

To design the EOP frame, we experimented with several distributions. When the mean of the EOP distribution deviates significantly from that of the latent, such as in $\mathcal{N}(1, 1)$, the model tends to generate strong noise near the end of the sequence. When the two distributions are numerically close, such as in $\mathcal{N}(0, 2)$, prediction becomes unreliable since the model struggles to recognize the stopping position. Moreover, because no KL constraint is applied to the latent, enforcing a specific probabilistic form for EOP further increases the modeling difficulty. Overall, we find that $\mathcal{N}(1, 0)$, i.e., a constant vector of ones, provides the most effective EOP representation, as it ensures clear numerical separation from latent features while remaining easy for the model to learn.

### 3.3.4 TRAINING AND INFERENCE PROCEDURES

The training and inference procedures are described in Algorithms 1 and 2. It is worth noting that the incorporation of the autoregressive mechanism allows us to enable KV cache at the block level during inference, which substantially accelerates the generation speed.

### 3.4 STOCHASTIC BLOCK REPA LOSS

As REPA loss is no longer required for lyric alignment constraints, we repurpose it to improve the model's musicality and structural modeling. For the target SSL representations, extracting them with the same block size as DiffRhythm 2 hinders the capture of larger-scale musical structures. Moreover, SSL representations are typically downsampled by multi-layer convolution, which often introduces several frame shifts. The misalignment between $z^i$ and SSL features caused by these shifts makes it unreliable to calculate REPA loss on target block directly. A better solution is to compute the loss on the entire sequence, which leverages the contextual receptive field to mitigate misalignment and enables the model to learn musical structures more effectively.

However, due to the block-wise training scheme, the hidden states $h_t^{i-1}$ and $h_t^i$ corresponding to $z_t^{i-1}$ and $z_t^i$ are not continuous, making it infeasible to directly compute REPA loss from the hidden state $H_t$ of $Z_t$. Considering that training is performed with teacher forcing, the hidden states $h^{<i}$ and $h_t^i$ corresponding to $z^{<i}$ and $z_t^i$ remain continuous. Thus, computing REPA loss on their combination is more reasonable. Note that for any $t$, the target is always the ground-truth SSL representation. Therefore, $t$ is only used to distinguish between noisy and clean sequences.

In practice, a noisy sequence typically contains dozens of blocks, so computing REPA loss for all of them is computationally expensive. Therefore, we randomly sample 10 blocks per noisy sequence for loss computation. For clean sequences, some blocks may be used multiple times for loss calculation. In such cases, we assign weights such that the total weight of each block sums to 1.

### 3.5 Cross-Pair Preference Optimization

Previous studies (Chen et al., 2025; Lei et al., 2025) have demonstrated the effectiveness and necessity of applying DPO for post-training in song generation tasks. As song generation involves multiple preference dimensions, multi-preference alignment optimization has become increasingly important. Existing approaches typically rely on merging separately optimized models. However, as the number of preferences increases, model merging often significantly reduces the average performance across preferences, severely limiting the overall efficiency of DPO.

We focus on four preference dimensions: musicality, style similarity, lyric alignment accuracy, and audio quality. Independent DPO experiments reveal complex interactions among these dimensions: improving lyric alignment often compromises musicality; optimizing audio quality may enhance alignment but weaken style similarity; while reinforcing style similarity tends to benefit musicality. These observations indicate that preferences can be either conflicting or synergistic.

Motivated by these findings, we propose a cross-pair preference optimization strategy. Preferences are grouped by similarity and paired across groups for pairwise optimization. Conflicting preferences are paired to mitigate trade-offs, while synergistic preferences are leveraged to enhance consistency across optimized models. This design not only reduces the number of optimized models required but also substantially improves the performance of the merged model.

In DiffRhythm 2, we pair (musicality, lyric alignment accuracy) and (style similarity, audio quality). During DPO training, the winning sample must satisfy both preferences within a pair, whereas the losing sample is required to satisfy at least one. To ensure balanced optimization, the three possible losing cases are preserved in equal proportions.

## 4 Experimental Setup

### 4.1 Dataset

DiffRhythm 2 is trained on a large-scale dataset comprising approximately 1.4 million songs, with a total duration of about 70,000 hours. The dataset spans three categories: Chinese, English and instrumental music, with a distribution ratio of roughly 4:5:1. To ensure data quality, we implement an efficient preprocessing pipeline. First, Audiobox-Aesthetics (Tjandra et al., 2025) is employed to filter audio based on quality. Then, Whisper (Cao et al., 2012) and FireRedASR (Xu et al., 2025b) are then used to transcribe the vocal tracks, and the two transcriptions are cross-validated against the original lyrics to ensure accuracy. Finally, All-in-One (Kim & Nam, 2023) and Qwen2.5-omni (Xu et al., 2025a) are used to annotate four key dimensions of the songs: structure, style, instrumentation, and emotion.

For evaluation, the test set consists of 50 real lyrics and 50 generated lyrics. For each lyric, we randomly select three style prompts, resulting in a total of 300 test cases. Across the entire test set, text prompts and audio prompts are balanced in equal proportion.

### 4.2 Evaluation Metrics

For the subjective evaluation, we invite ten listeners with professional music backgrounds to provide Mean Opinion Score (MOS) score. Each generated song is rated along four dimensions: musicality (MUS), harmony between vocals and accompaniment (HAR), vocal performance (VOC), accompaniment performance (ACC), and overall performance (OVP).

For the objective evaluation, four aspects are considered: music quality, audio quality, style similarity, and lyric accuracy. To evaluate lyric accuracy, we transcribe the generated songs using Qwen3 ASR [2] and compute the phoneme error rate (PER). Style similarity is evaluated with MuQ-Mulan (Zhu et al., 2025), which measures the similarity of both textual prompts (Mulan-T) and audio prompts (Mulan-A). Music quality is assessed using SongEval (Yao et al., 2025) in terms of overall coherence (CO), memorability (ME), naturalness of vocal breathing and phrasing (NA), clarity of song structure (CL), and overall musicality (MU). Audio quality is evaluated with Audiobox-Aesthetics (Tjandra et al., 2025), which considers content enjoyment (CE), content usefulness (CU),

---

[2] https://qwen.ai/blog?id=e199227023e8ebaac5f348f97fa804d1858ffc8a&from=research.research-list

Table 1: Results of objective evaluations. The best results among commercial models are highlighted in bold, while the best results among open-source models are underlined.

| Model | PER ↓ | Mulan-T ↑ | Mulan-A ↑ | Audio Aesthetics ↑ | | | | SongEval ↑ | | | | |
|---|---|---|---|---|---|---|---|---|---|---|---|---|
| | | | | CE | CU | PC | PQ | CO | MU | ME | CL | NA |
| SUNO V4.5 | 0.28 | **0.38** | - | **7.78** | **7.85** | 6.28 | **8.44** | **4.27** | 4.01 | 4.04 | **3.98** | **3.89** |
| Mureka-O1 | 0.09 | 0.37 | - | 7.65 | 7.81 | **6.31** | 8.35 | 4.14 | **4.06** | **4.07** | 3.93 | 3.85 |
| DiffRhythm+ | 0.15 | 0.25 | 0.69 | 7.44 | 7.51 | 6.22 | 7.85 | 3.63 | 3.39 | 3.42 | 3.61 | 3.15 |
| ACE-Step | 0.23 | 0.28 | - | 7.26 | 7.51 | 6.25 | 7.79 | 3.77 | 3.46 | 3.58 | 3.56 | 3.38 |
| LeVo | 0.19 | 0.35 | 0.81 | 7.51 | 7.78 | 5.68 | 8.12 | 3.74 | 3.56 | 3.62 | 3.55 | 3.40 |
| DiffRhythm 2 | 0.13 | 0.40 | 0.75 | 7.48 | 7.59 | 6.12 | 7.91 | 4.09 | 3.93 | 4.01 | 3.89 | 3.78 |

Table 2: Results of subjective evaluations. The best results among commercial models are highlighted in bold, while the best results among open-source models are underlined.

| Model | MUS ↑ | HAR ↑ | VOC ↑ | ACC ↑ | OVP ↑ |
|---|---|---|---|---|---|
| SUNO V4.5 | 3.68 | **4.03** | 3.61 | **3.79** | **3.92** |
| Mureka-O1 | **3.71** | 3.99 | **3.63** | 3.70 | 3.87 |
| DiffRhythm+ | 3.10 | 3.22 | 2.91 | 3.42 | 3.27 |
| ACE-Step | 3.40 | 3.75 | 3.38 | 3.60 | 3.55 |
| LeVo | 3.48 | 3.68 | 3.46 | 3.27 | 3.56 |
| DiffRhythm 2 | 3.57 | 3.81 | 3.31 | 3.64 | 3.77 |

production complexity (PC), and production quality (PQ). We further explore the generation speed in Appendix B.

## 4.3 COMPARISON SYSTEMS

We conduct a comprehensive comparison of DiffRhythm 2 against multiple systems. For benchmarking, we select two industry-leading commercial systems: Suno V4.5 [3] and Mureka-O1 [4]. In addition, we include three open-source systems for evaluation: DiffRhythm+ [5], ACE-Step [6], and LeVo [7].

## 5 EVALUATION RESULTS

## 5.1 OBJECTIVE RESULTS

From Table 1, DiffRhythm 2 significantly outperforms other open-source models across music quality metrics. However, in aspects such as musicality, it still shows a clear gap compared to commercial systems like SUNO V4.5. In terms of audio quality, it performs slightly worse than LeVo but remains superior to most other models. This result demonstrates that employing a lower frame rate is feasible for compressed song reconstruction, although the increased reconstruction burden makes it difficult to achieve perfect fidelity. DiffRhythm 2 achieves the highest scores on Mulan-T (0.40) and PER (0.13), substantially exceeding other models. However, its Mulan-A score is lower than that of LeVo and similar systems, which we attribute to our choice of modeling audio style prompts with global representations. A global style embedding cannot adequately capture the stylistic content present in songs.

---

[3]https://suno.com/blog/introducing-v4-5

[4]https://www.mureka.ai

[5]DiffRhythm+ is tested using DiffRhythm+-full released at https://github.com/ASLP-lab/DiffRhythm

[6]ACE-Step is tested using the code released at https://github.com/ace-step/ACE-Step

[7]LeVo is tested using the code released at https://github.com/tencent-ailab/songgeneration

Table 3: Results of ablation study on DiffRhythm 2.

| Model | PER ↓ | Mulan-T ↑ | Mulan-A ↑ | Audio Aesthetics ↑ | | | | SongEval ↑ | | | | |
|---|---|---|---|---|---|---|---|---|---|---|---|---|
| | | | | CE | CU | PC | PQ | CO | MU | ME | CL | NA |
| DiffRhythm 2 | 0.13 | 0.40 | 0.75 | 7.48 | 7.59 | 6.12 | 7.91 | 4.10 | 3.93 | 4.01 | 3.89 | 3.78 |
| w/o DPO | 0.27 | 0.34 | 0.70 | 7.24 | 7.56 | 5.46 | 7.81 | 3.67 | 3.46 | 3.53 | 3.39 | 3.33 |
| w/o CPPO | 0.18 | 0.37 | 0.69 | 7.27 | 7.64 | 5.68 | 7.82 | 3.79 | 3.57 | 3.65 | 3.50 | 3.43 |
| w/o REPA | 0.15 | 0.30 | 0.68 | 7.13 | 7.37 | 6.01 | 7.72 | 3.92 | 3.73 | 3.82 | 3.65 | 3.61 |

## 5.2 Subjective Results

As shown in Table 2, DiffRhythm 2 performs notably better than other open-source models in musicality, vocal-accompaniment harmony, and overall quality. Its accompaniment performance is comparable to that of ACE-Step and far exceeds that of LeVo, demonstrating that continuous representations can effectively enhance accompaniment generation. However, in vocal quality, since our system neither leverages semantic constraints nor separately models the vocal track, it falls slightly behind ACE-Step and LeVo. Overall, though, commercial models still maintain a clear advantage over open-source systems.

## 5.3 Ablation Study

To validate the effectiveness of our proposed methods, we conduct ablation studies focusing on stochastic block REPA loss and cross-pair preference optimization. For the cross-pair preference optimization ablation (w/o CPPO), we replaced it with separate DPO on four preferences followed by model merging. For the stochastic block REPA loss ablation (w/o REPA), we simply removed the corresponding loss during training. In addition, as a comparison, we included an experiment without applying any DPO (w/o DPO).

As shown in Table 3, removing REPA loss leads to a clear drop in SongEval scores. Moreover, listening tests reveal that music structure become noticeably misaligned or even fail entirely. Similarly, without cross-pair preference optimization, all evaluation metrics decline significantly compared to DiffRhythm 2.

## 6 Conclusion and Discussion

In this paper, we present DiffRhythm 2, a semi-autoregressive end-to-end music generation framework based on block flow matching. By leveraging this approach, we achieve high-quality lyric alignment without relying on semantic constraints, while generating mixed-track songs with high fidelity. We further introduce stochastic block REPA loss for semi-autoregressive training, which enhances musicality and structural modeling. In addition, we propose cross-pair preference optimization, which effectively addresses the challenge of degraded average performance when optimizing across multiple preferences. Our experiments demonstrate the superior song generation capabilities of DiffRhythm 2.

However, experiments also reveal certain limitations. The low-frame-rate VAE imposes an upper bound on the fidelity of reconstructed audio, making it difficult to match real audio quality. Furthermore, improving vocal modeling without compromising model creativity remains a key challenge for enhancing mixed-track generation. More broadly, it is evident that open-source models still fall short of commercial systems in overall performance, necessitating for further advancements in data and generation strategies.

As DiffRhythm 2 is capable of generating complete songs, it could potentially be misused to create disinformation, deepfake audio, or other harmful content. Being committed to the responsible advancement of this field, we will provide appropriate usage restrictions and guidelines when releasing the open-source code and model checkpoints.

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

## A    RESULTS OF THE MUSIC RECONSTRUCTION

We evaluated the compression and reconstruction performance of Stable Audio 2 VAE, the DCAE from ACE-Step, the dual-track MuCodec from LeVo and our implemented Music VAE on a testset of 50 songs covering diverse genres. It should also be noted that Music VAE supports only mono audio, whereas all the other models support stereo audio. As shown in the table, our Music VAE achieves comparable or even higher PESQ and STOI scores despite operating at a substantially lower frame rate than the other models. However, the WER of Music VAE is slightly higher than other models, indicating that the extreme low frame rate still results in some loss of fine-grained reconstruction details. The single codebook design of MuCodec significantly compromises the quality of the accompaniment and leads to inferior overall reconstruction performance.

By examining the outputs of the generative model, we find that the reconstruction quality of generated audio is generally superior to that of audio obtained through direct encoding and decoding with codec or VAE. In reconstruction tests, issues such as pronounced pronunciation errors or blurred accompaniment frequently occur, whereas their occurrence is greatly reduced in generated audio. Consequently, objective metrics based solely on codec or VAE reconstructions may fail to accurately reflect the quality achievable by the generative model.

Table 4: Comparison of reconstruction performance across different music compression models

| Model | Frame Rate | Sample Rate | PESQ ↑ | STOI ↑ | PER ↓ |
|---|---|---|---|---|---|
| Stable Audio 2 VAE | 21.5Hz | 44100Hz | 1.981 | 0.634 | 0.148 |
| DCAE (ACE-Step) | 10Hz | 44100Hz | 2.176 | 0.647 | 0.117 |
| MuCodec (Dual-Track) | 25Hz | 48000Hz | 1.876 | 0.561 | 0.174 |
| Music VAE | 5Hz | 48000Hz | 2.477 | 0.683 | 0.121 |

## B    GENERATION SPEED

We compare the generation speed of LeVo, Yue, DiffRhythm+, and DiffRhythm 2. To ensure fairness, all models are required to generate a fixed sequence of two minutes in length, regardless of generation quality. For both DiffRhythm+ and DiffRhythm 2, the number of sampling steps is set to 32. As shown in Table 5, DiffRhythm 2 is only slightly slower than DiffRhythm+ and ACE-Step, while being significantly faster than LeVo. It is worth noting that DiffRhythm+ does not incorporate any attention acceleration framework, which explains why it is slower than ACE-Step despite having a smaller model size. ACE-Step leverages linear attention to reduce computational complexity, whereas LeVo adopts Flash Attention [8], and DiffRhythm 2 employs Flex Attention [9] for acceleration.

Table 5: Generation speed comparison on RTX 4090.

| Model | Architecture | Model Size | Time Cost ↓ | RTF ↓ |
|---|---|---|---|---|
| DiffRhythm+ | Flow Matching + VAE Decoder | 1B + 150M | 18.3s | 0.153 |
| ACE-Step | Flow Matching + DACE + Vocoder | 3.5B + 150M | 15.2s | 0.127 |
| LeVo | LM + Diffusion Decoder | 2B + 0.7B | 147s | 1.225 |
| DiffRhythm 2 | Block Flow Matching + Music VAE Decoder | 1B + 170M | 25.6s | 0.213 |

---

[8]https://github.com/Dao-AILab/flash-attention
[9]https://pytorch.ac.cn/blog/flexattention/

# C  TRAINING DETAILS

Table 6: Music VAE training details.

| Parameter | Specification |
| --- | --- |
| Dataset | 70,000-hour music dataset and 100,000-hour speech dataset. |
| Latent Space | Encoder produces a latent representation with a latent dimension of 64 frames. |
| Hardware | 16 NVIDIA A100 GPUs. |
| Training Steps | 1,500,000 steps. |
| Batch Size (Global) | 128 (8 per GPU). |
| Total Duration | Approx. 7 days. |

Table 7: DiffRhythm 2 training details.

| Parameter | Specification |
| --- | --- |
| Model Size | Approx. 1B |
| Computational Resources | 32 NVIDIA A100 GPUs. |
| Batch Size | 64 (2 per GPU). |
| Block Size | 10 |
| Steps | 1,000,000 steps for pretraining; 200,000 steps for finetuneing; 2 epochs for DPO. |
| Dataset | 70,000-hour music dataset for pretrain; high-quality subset of 20,000 hours for fintune; 40,000 pairs for DPO. |
| Total Training Time | 200 hours. |
| Optimizer Type | AdamW with 1e-2 weight decay and (0.8, 0.9) betas. |
| Gradient Clipping | Max norm of 0.5. |
| Learning Rate | 1e-4, with linear warm-up over the first 10,000 steps; 1e-5 for finetuning. |
| Timestep Sampling | Uniform scheme: $t \sim \mathcal{U}[0, 1]$. |

