# OpenReview forum: "DiffRhythm 2: Efficient and High Fidelity Song Generation via Block Flow Matching"
_ICLR.cc/2026/Conference — Submitted to ICLR 2026_

### Official Review · Reviewer_Ph3U · 2025-10-27

**Soundness:** 3
**Presentation:** 3
**Contribution:** 2
**Rating:** 4
**Confidence:** 4

**Summary:**

This technical paper proposes DiffRhythm 2, a chunk-based diffusion autoregressive model for full song generation.
The main design choices of the presented approach are:
- using autoregressive chunk-wise generation, where each chunk is modeled using Flow matching, conditioned on the previous clean samples. Not novel, very similar to "Block Diffusion: Interpolating Between Autoregressive and Diffusion Language Models" in the text domain.
- leveraging Representation alignment loss (REPA) as an extra regularization. To allow the use of pretrained feature extractors trained on longer audio chunks (here MuQ), noisy features are concatenated with clean features to produce sequences of suitable size. Since it's too costly to regularize all such subsequences, only a few are kept, leading to the proposed "stochastic block REPA loss"
- "cross-pair preference optimization strategy", which only consists in slight modification of how samples are ranked during the DPO phase: "we pair (musicality, lyric alignment accuracy) and (style similarity, audio quality). During DPO training, the winning sample must satisfy both preferences within a pair, whereas the losing sample is required to satisfy at least one".
- training over 1.7million songs

If not necessarily novel, these design choices are carefully evaluated using objective and subjective evaluations. These reveal consistent improvements over the previous iteration DiffRhythm+ and other open source music generation models, while still staying below commercial models like SUNO v4.5.

**Strengths:**

The plan to release code and checkpoints is an important point but needs to effectively done, as this would be the main contribution of this paper.
The design choices are relevant and proved to be effective enhancing the quality over previously released open source models.
If effective and if the accompanying website features interesting generation, this paper is still rather incremental.

**Weaknesses:**

The writing is rather general and we can regret that some technical points are not referenced.
e.g. "SSL representations are typically downsampled by multi-layer convolution, which often introduces several frame shifts."
"Existing approaches typically rely on merging separately optimized models."

Some repetitions between intro and related works.
Lack of references beyond music generation. Many building blocks already exists in the literature and should be properly cited like
- Block Diffusion
- Diffusion DPO
since they are used in the presented approach.
Overall, the paper is not really self-contained: a more detailed paragraph on REPA could help. The reader needs to check in the overview section 3.1 to know what SSL backbone was used (there's no mention of it in the main 3.4 STOCHASTIC BLOCK REPA LOSS section).
It is also hard to understand the DPO procedure from the paper alone.

**Questions:**

Can you detail the DPO training? it seems that it should be some sort of DiffusionDPO, but the paper is not cited.

**Details Of Ethics Concerns:**

Unclear regarding data sources:
"DiffRhythm 2 is trained on a large-scale dataset comprising approximately 1.4 million songs, with
a total duration of about 70,000 hours"

---

### Official Review · Reviewer_pj94 · 2025-10-30

**Soundness:** 3
**Presentation:** 2
**Contribution:** 3
**Rating:** 4
**Confidence:** 4

**Summary:**

This paper proposes DiffRhythm 2, a semi-autoregressive song generation framework based on block flow matching to improve alignment between lyrics and vocals without relying on timestamp labels. The model integrates a Music VAE with a 5 Hz low-frame-rate compression, which enables efficient long-sequence modeling, and introduces stochastic block REPA loss to enhance structural coherence and musicality. Furthermore, a cross-pair preference optimization strategy is proposed to mitigate conflicts among multiple human-preference dimensions.

**Strengths:**

1. The block flow matching addresses lyric–vocal alignment without timestamp labels, reducing data preprocessing requirements and improving usability.
2. The cross-pair preference optimization takes into account the interdependence among different optimization dimensions in song generation.
3. The work involves substantial engineering effort and system implementation.

**Weaknesses:**

1. The definition of “lyrics alignment” is insufficiently precise. It is unclear whether this term refers purely to lyric accuracy (i.e., no omissions or mispronunciations of words) or also encompasses prosodic naturalness—such as rhythm, phrasing, and pauses. In general, compared with AR models, non-autoregressive (NAR) models tend to excel in lyric accuracy but struggle with natural rhythmic expressiveness.
2. Block flow matching is not a new idea; it has been applied in video generation [1] and text-to-speech synthesis [2]. Please include proper citations in the method section and clarify what methodological novelty this paper introduces beyond these prior works.
3. Could you provide the comparison results of generation quality and speed with Yue? In the appendix, the paper states, “We compare the generation speed of LeVo, Yue, DiffRhythm+, and DiffRhythm 2,” but the results for Yue are not included.
4. The presentation quality could be improved. Figures 1 and 2 convey limited information, and Sections 3.4 and 3.5 would benefit from additional diagrams to support the textual descriptions.

[1] Zhang, Yuan, et al. Generative Pre-trained Autoregressive Diffusion Transformer. arXiv:2505.07344 (2025).
[2] Liu, Zhijun, et al. Autoregressive Diffusion Transformer for Text-to-Speech Synthesis. arXiv:2406.05551 (2024).

**Questions:**

1. What are the specific innovations when applying block flow matching to the text-to-song generation task?
2. How was the block size determined, and how does it affect model performance and stability?
3. The paper mentions 70,000 hours of music data and 100,000 hours of speech data. How are these two datasets specifically used during training? In the first example, compared with LeVo, the generated songs show less natural rhythm, weaker vocal techniques, and a relatively thin accompaniment. Does this mean that the speech data had a stronger influence on the model?
4. Are there controlled ablation experiments verifying the contribution of block flow matching to lyric accuracy and rhythmic naturalness, while excluding the effects of different training datasets?
5. The paper introduces a 5 Hz Music VAE. What enables such a low bitrate? You report reconstruction metrics, but how does this low frame rate affect generation quality? What would happen if a higher-bitrate VAE were used?
6. How was the evaluation set constructed? What are its data sources and style distributions? How do you ensure fairness across systems, and will this dataset be released publicly?

**Details Of Ethics Concerns:**

The paper presents a full-song generation model that can synthesize realistic human singing voices. Such capability raises potential misuse concerns, including impersonation, deepfake generation, and copyright infringement. Clear ethical guidelines and data usage disclosures are recommended.

---

### Official Review · Reviewer_VLPj · 2025-11-01

**Soundness:** 3
**Presentation:** 3
**Contribution:** 3
**Rating:** 8
**Confidence:** 4

**Summary:**

This paper introduces DiffRhythm 2, a semi-autoregressive song generation framework built on block flow matching. The system (i) compresses audio with a 5 Hz Music VAE to keep long sequences tractable; (ii) achieves lyrics–vocal alignment without external timestamps via block-wise flow matching and an EOP (end-of-prediction) mechanism; (iii) improves musical structure using a stochastic block REPA loss; and (iv) addresses multi-preference post-training with cross-pair preference optimization (CPPO) rather than model merging. Experiments on ~300 prompts report better objective/subjective scores than prior open-source systems (DiffRhythm+, ACE-Step, LeVo), competitive generation speed, and up to 210 s coherent songs. The paper positions itself as a fast, high-fidelity, controllable text/audio-to-song generator.

**Strengths:**

1. Clear technical framing & originality. The block flow matching formulation that is non-AR within block yet AR across blocks is a neat, minimalistic way to get alignment while keeping efficiency. The timestep trick (S/L set to −1, clean=1, noisy∈[0,1]) to disambiguate streams is simple and effective.

2. Practical long-sequence engineering. Using a 5 Hz VAE plus block-level KV cache gives a realistic path to multi-minute songs with stable inference time; the paper also discusses the EOP design choices and their failure modes.

3. PPO avoids the usual degradation from weight-space interpolation of multiple DPO heads.

4. Good empirical results. Objective metrics (PER, Mulan-T/A, Audiobox-Aesthetics, SongEval) and ablation (w/o DPO, w/o CPPO, w/o REPA) are comprehensive; results are consistently strong vs. open-source baselines with a reasonable gap to top commercial systems.

**Weaknesses:**

1. My main concern is that this paper reads more like a technical report or system description rather than a research paper with focused insights.

Although the engineering contributions like block flow matching, low-frame-rate VAE, stochastic REPA loss, and cross-pair preference optimization are each well-motivated and empirically validated, the work overall feels like a composition of effective engineering tricks rather than a cohesive theoretical or methodological advancement.

2. Missing baselines: SongGen, Yue, and SongBloom.

**Questions:**

1.  While PESQ (perceptual evaluation of speech quality) and STOI are widely used for speech quality,  is it reasonable to use the same metrics for music ?

2.  Another issue is that the choice of the Music VAE is not sufficiently justified. Because it is well known that good reconstruction does not necessarily imply good generation quality [1].

[1] Yao J, Yang B, Wang X. Reconstruction vs. generation: Taming optimization dilemma in latent diffusion models[C]//Proceedings of the Computer Vision and Pattern Recognition Conference. 2025: 15703-15712.

---

### Official Review · Reviewer_UTAb · 2025-11-03

**Soundness:** 3
**Presentation:** 3
**Contribution:** 3
**Rating:** 4
**Confidence:** 5

**Summary:**

This paper introduces DiffRhythm2, a method for full-song generation based on block flow matching. The approach models music signals using a variational autoencoder (VAE) that operates at a low frame rate of 5 Hz. The authors further propose a cross-pair preference optimization strategy to enhance robustness across diverse human preferences, complemented by a musicality and structural coherence alignment loss. Experimental results show that DiffRhythm2 is capable of generating complete songs of up to 210 seconds, outperforming existing open-source models in both subjective and objective evaluations.

**Strengths:**

1. The method is simple, elegant, and easy to follow
2. The paper is well written and easy to follow
3. Results and provided samples are impressive

**Weaknesses:**

1.	The experimental setup could be strengthened, particularly in relation to the paper’s primary contribution.
2.	Several methodological details and results are missing—for example, information regarding the VAE configuration, hyperparameters, and related implementation choices.
3.	The overall contribution of the proposed approach could be better articulated and justified. At the moment, some components appear to be loosely integrated ideas aimed primarily at improving generation performance, rather than forming a cohesive methodological framework.

**Questions:**

1.	The authors describe training a VAE with various reconstruction losses and discriminators. However, it is unclear whether any KL regularization or other latent constraints were applied. In other words, what specifically distinguishes this model as a variational autoencoder rather than a standard autoencoder? Based on the current description, it appears closer to an AE than a true VAE.
2.	In Table 2, the authors report subjective evaluation results. It would be helpful to include measures of variability—such as standard deviation or 95% confidence intervals—to better assess the statistical significance of these findings.
3.	Table 3 presents ablation studies over different loss functions, which is valuable. However, additional ablations on the model architecture would further strengthen the analysis—especially since architectural design is highlighted as a primary contribution and emphasized in the title. For example, what are the advantages of using block diffusion? how does the block size influence performance? etc.
4.	The related work section should include appropriate citations. For instance, the statement “Conditional flow matching (CFM) further conditions the vector field on external information, enabling guided generation. CFM is widely used in the field of image generation and has also been applied to TTS tasks in recent years.” requires supporting references to prior work.

---

> ### Author Response · Authors · 2025-11-13
>
> Q: Experimental setup issue
>
> A: We will refine the experimental setup in the revision to present a clearer and stronger validation of the proposed framework.
>
> Q: The configuration of VAE and the loss of VAE
>
> A: The VAE encoder has a total downsampling ratio of 4800 and contains approximately 100M parameters. The transformer block within the VAE consists of four transformer decoder layers, totaling around 50M parameters. The decoder adopts the BigVGAN architecture with an upsampling ratio of 4800 and about 150M parameters.
> Our VAE includes KL loss, reconstruction loss and discriminator loss, which makes it a true VAE rather than a simple AE. These details will be included in the revised paper.
>
> Q: Some components appear to be loosely integrated ideas aimed primarily at improving generation performance, rather than forming a cohesive methodological framework.
>
> A:
> Thank you for the insightful feedback. DiffRhythm 2 is not a loose collection of tricks, but a tightly integrated framework built around a semi-autoregressive block-based generation paradigm.
> - Block Flow Matching forms the core, enabling long-sequence modeling with natural lyric alignment.
> - The High-Compression Music VAE is a key enabler, making block-level autoregressive training computationally feasible.
> All components are designed synergistically to solve the fundamental challenges in song generation: alignment, coherence, multi-preference balancing, and efficiency, within a unified and scalable architecture. We will revise the paper to better articulate this cohesive narrative and highlight how the components form an integrated methodology rather than isolated ideas.
>
> Q: The 95% confidence intervals of Table 2
>
> A:
>
> | Model        | MUS ↑         | HAR ↑         | VOC ↑         | ACC ↑         | OVP ↑         |
> | ------------ | ------------- | ------------- | ------------- | ------------- | ------------- |
> | SUNO V4.5    | 3.68±0.14     | **4.03±0.11** | 3.61±0.12     | **3.79±0.15** | **3.92±0.10** |
> | Mureka-O1    | **3.71±0.12** | 3.99±0.15     | **3.63±0.13** | 3.70±0.11     | 3.87±0.11     |
> | DiffRhythm+  | 3.10±0.21     | 3.22±0.20     | 2.91±0.17     | 3.42±0.19     | 3.27±0.24     |
> | ACE-Step     | 3.40±0.18     | 3.75±0.18     | 3.38±0.19     | 3.60±0.17     | 3.55±0.18     |
> | LeVo         | 3.48±0.22     | 3.68±0.19     | *3.46±0.14*   | 3.27±0.27     | 3.56±0.20     |
> | DiffRhythm 2 | *3.57±0.20*   | *3.81±0.18*   | 3.31±0.19     | *3.64±0.16*   | *3.77±0.18*   |
>
> Q: Advantage of Block Diffusion and the influence of different block size
>
> A:
> The advantage of Block Diffusion lies in its ability to better predict continuous representations [1] while addressing the alignment issues in long sequences.
>  Different block sizes primarily affect the phoneme error rate (PER) and generation speed, with minimal impact on musicality and other perceptual aspects. Smaller block sizes result in lower PER and more stable alignment but slower generation, making the overall behavior closer to that of an autoregressive model. In contrast, larger block sizes can lead to more skipped or missing tokens, with overall behavior resembling a non-autoregressive model.
> | Block Size | PER  | RTF  |
> | -----| ----- | ----- |
> | 5 | 0.11  | 0.455 |
> | 10 | 0.13  | 0.213 |
> | 20 | 0.17  | 0.154 |
> | 100 | 0.23  | 0.176 |
>
> Q: Appropriate citations.
>
> A: We thank the reviewer for pointing this out. We agree that the statements regarding CFM should be supported with appropriate citations. In the revised manuscript, we will add relevant references to prior works in both image generation and TTS to properly support these

---

### Meta-Review · Area_Chair_aPmm · 2025-12-27

**Summary:**

The reviewers raised substantial concerns about the paper’s novelty and missing implementation details, and these issues were not adequately addressed in the rebuttal. I therefore recommend rejecting the paper.

Concerns on Novelty: **UTAb, VLPj, pj94** expressed concern on the novelty of the paper. For example, **pj94** mentioned that Block flow matching has been applied in video generation and text-to-speech synthesis.  **VLPj** mentioned that the paper feels like a composition of effective engineering tricks rather than a cohesive theoretical or methodological advancement.

Concerns on Missing Implementation Details: **UTAb,** **Ph3U** and **pj94** have concerns on missing implementation details about VAE hyperparameters (e.g., **pj94** questioned about details on the 5 Hz Music VAE) and the DPO procedure.

Concerns on Experiments: **VLPj** and **pj94** both think the author needs to compare with Yue (**VLPj** also mentioned SongGen and SongBloom). **pj94** raised further concerns on lack of experiments that show the contribution of block flow matching to lyric accuracy and rhythmic naturalness and the lack of details about the evaluation set.

Concerns on Presentation Quality: **UTAb, pj94 and Ph3U** all share the concern on presentation quality of the paper, e.g., justification of the approach, information included in the figures, and reference to related work.

**Reviewer Concerns:**

The authors provide additional details about the VAE in the rebuttal, but these clarifications are still insufficient to fully address the reviewers’ concerns. On the other hand, the added 95% confidence intervals and the new results analyzing how block size affects performance do help address earlier questions about experimental design and hyper-parameter choices.

**Reviewer Scores:**

I think the reviewers will keep the score because the author did not address most of the concerns.

---

### Decision · Program_Chairs · 2026-01-26

Reject